# The *Solanum torvum* Transcription Factor *StoWRKY6* Mediates Resistance against Verticillium Wilt

**Yu Zhang, Lei Shen, Liangjun Li and Xu Yang \***

College of Horticulture and Plant Protection of Yangzhou University, 88 South Daxue Road, Yangzhou 225009, China

\* Correspondence: yangxu@yzu.edu.cn

**Abstract:** WRKY is a transcription factor family that has attracted much attention in recent studies of plant disease resistance, but there are few reports in the study of eggplant resistance to Verticillium wilt. Here, we retrieved an up-regulated *WRKY* transcription factor, *StoWRKY6*, from the transcriptome sequencing data of *Solanum torvum* response to *Verticillium dahliae* infection. Phylogenetic analyses revealed the highest homology species of *StoWRKY6* in the WRKY family is *Solanum melongena*. Based on the quantitative real-time PCR analysis, *StoWRKY6* was highly expressed in the roots but barely expressed in the leaves. Transient expressions of *StoWRKY6* in *Nicotiana benthamiana* showed a nuclear localization. A virus-mediated gene silencing experiment indicated that the silencing of *StoWRKY6* reduced the resistance to Verticillium wilt in *Solanum torvum*. To further verify the immune response function, we introduced *StoWRKY6* into *Nicotiana benthamiana* using transient transformation technology and found obvious spots under UV light. In summary, these results showed that *StoWRKY6* played an important role in the resistance to Verticillium wilt of *Solanum torvum*, which may function mainly by inducing an immune response. Our study provided strong evidence for the mechanism of eggplant resistance to Verticillium wilt and laid a foundation for the potential molecular breeding of eggplant disease resistance.

**Keywords:** *Solanum torvum*; Verticillium Wilt; *StoWRKY6*; defense response

## 1. Introduction

Eggplant (*Solanum melongena*) is one of the most important vegetable crops in the world, and the eggplant Verticillium wilt, which is caused by the soil-borne fungus *Verticillium dahliae* kleb., is an important vascular disease that seriously reduces the yield and quality of eggplant. This disease has been reported in many countries in the world [1]. Verticillium wilt is extremely difficult to control in production and the most economical and most effective measure to reduce losses is to plant disease-resistant varieties. At present, many studies have been conducted on the resistance mechanism of eggplant to verticillium wilt. However, the research on the resistance mechanism remains focused on cloning and quantitative expression of individual differentially expressed genes. The study of gene function and the analysis of disease resistance mechanisms need to be further deepened.

Plants often suffer from various biotic and abiotic stresses during their growth, and the expression of stress-induced genes at the transcription level plays a crucial role in regulating the level of protein by transcription factors (TFs). The WRKY family is one of the largest transcription factor families in plants and many of its members in different plant species have been cloned. The model plant *Arabidopsis thaliana* (Arabidopsis) has 74 members [2], rice (*Oryza sativa*) has more than 100 members [3], soybean (*Glycine max*) has 197 members [4], and tomato (*Solanum lycopersicum*) has 81 members [5] belonging to the WRKY family. The name of the WRKY transcription factor is derived from its highly conserved N-terminal WRKY domain consisting of 60 amino acids [6,7]. The type of zinc finger motifs and the number of WRKY domains jointly determine the classification of

WRKY proteins. According to these characteristics, the WRKY can be divided into three categories. Group I WRKYs contain two WRKY domains, whereas Group II and III WRKYs contain only one WRKY domain. In addition, WRKYs belonging to Group I and Group II have a $C_2H_2$ zinc finger motif, whereas Group III WRKYs have a C2HC zinc finger motif.

Studies have shown that WRKY transcription factors bind with the W-box and interact with it specifically via its conserved amino acid sequence [8]. WRKY transcription factors bind with the W-box of the target gene promoter to regulate the expression of target genes involved in a variety of biological and metabolic reaction processes in plants. Studies have shown that WRK transcription factors are not constitutively expressed in plants but are induced by various stresses. AtWRKY33 is induced by pathogens to enhance the resistance of plants against *Alternaria* and *Botrytis cinerea* [9,10]. *CaWRKY27* transcript levels are up-regulated by treatments with salicylic acid (SA), methyl jasmonate (MeJA), and ethephon (ETH). Transgenic tobacco plants overexpressing *CaWRKY27* exhibit resistance to *R. solanacearum* infection compared to wild-type plants [11]. The expression of *SlDRW1* is significantly induced by *B. cinereal* with a 10 to 13-fold increase compared to that in mock-inoculated plants. The silencing of SlDRW1 results in increased severity of disease caused by *B.cinerea* [12]. Silencing *SlWRKY70* attenuated Mi-1-mediated resistance against both potato aphid and RKN, suggesting that *SlWRKY70* is required for the function of *Mi-1* [13]. Cloning, isolation, and functional analysis of WRKYs have become a hot topic in the field of plant molecular biology. Plenty of studies have been carried out on non-solanaceous crops, such as rice, Arabidopsis, and Solanaceae tomato, whereas research on other Solanaceae plants is relatively sparse. Therefore, the research of Solanaceae WRKY transcription factors is conducive to the further understanding of the physiological and biochemical responses of Solanaceous plants under stress conditions.

WRKY6 has been reported to be involved in many growth, development, and stress response processes. The first reported function of WRKY6 is to participate in plant pathogen defense and the aging process. Studies have shown that WRKY6 expression is significantly increased in aging leaves and bacterial infection, and WRKY6 can positively activate the PR1 promoter [14,15]. In addition, WRKY6 is classified as a direct early type gene that does not require de novo synthesis of protein activating enzymes; it is most likely that this gene is involved in regulating some early steps of these processes [16].

Robatzek [15] found that the ectopic over-expression of WRKY6 affects the promoter activity of PR1 by positively regulating the transcription factor NPR1, indicating that WRKY6 plays a role upstream of NPR1 and PR1 in the general stress response pathway. The transcriptional regulatory responses of PR1 and WRKY6 are similar, for example, in continuous β-aminobutyric acid induction; additionally, when treated with salicylic acid, WRKY6 and PR1 showed enhanced expression levels [17], indicating that plant hormones can directly or indirectly regulate the expression of the WRKY transcription factor and PR protein, further indicating the important role of both in the plant disease resistance response. Currently, there are few studies on WRKY transcription factors in eggplant. Yang [18] identified 58 genes encoding WRKY protein in eggplant by a genome-wide search and found WRKY genes differentially expressed under low-temperature treatment through RNA-seq. In addition, Shao has cloned and analyzed the basic sequence of eggplant SmWRKY1, but the major function has not been thoroughly explored [19]. The function of WRKY TFs in eggplant needs to be further studied. We screened and cloned a wild eggplant *Solanum torvum* transcription factor WRKY gene in this study, *StoWRKY6*. Quantitative analysis of the expression of the gene in the wild in different tissues and the quantitative detection of *Solanum torvum* regarding expression changes before and after infection with *Verticillium dahliae* were conducted. The subcellular localization of StoWRKY6 was examined and the expression pattern was observed by transient expression of the gene in tobacco. Furthermore, the function was verified by inoculation with pathogenic *Verticillium dahliae*; therefore, we provide a theoretical basis for breeding new varieties of disease resistant eggplant.

## 2. Materials and Methods

### 2.1. Materials and Inoculation Method

*Solanum torvum* and *Nicotiana benthamiana* plants were grown in a mixture of perlite:vermiculite:plant ash (1:6:2) in a growth chamber at 22 °C under the 12 h-light and 12 h-dark cycle condition.

The *Verticillium dahliae* was resuscitated on a potato dextrose agar plate at 25 °C and cultured in Czapek liquid in a swing bed. After 7 days, the spores of *Verticillium dahliaee* were harvested by filtering through eight layers of gauze and subsequently counted by hemocytometer. For *Verticillium dahliaee* inoculation, plants of *Solanum torvum* with four true leaves were uprooted, dipped for 5 min in a suspension containing $1 \times 10^7$ spores/mL, and replanted in vermiculites.When inoculating, carefully take out the water eggplant seedlings with basically the same growth from the substrate, wash the roots with tap water, place them in a triangular flask, add the above mixture to the cotyledon, and seal the bottle mouth with a sealing film. The inoculated seedlings were cultured in a constant temperature incubator at 25 °C for 12 h/12 h light/dark, and the light intensity was 20,000 LX. After0, 12, 24, 36, 48 and 72 h after infection, the roots of the plants were cut off, washed with sterile distilled water, quickly frozen in liquid nitrogen for 10 min, and stored in an ultra-low temperature refrigerator at −80 °C. Each treatment was repeated 3 times.

### 2.2. Extraction of Total RNA and Genome DNA

The ultra-low-temperature frozen plant tissue samples were quickly transferred to a mortar precooled with liquid nitrogen, and the tissues were ground with a pestle, and liquid nitrogen was continuously added until they were ground into powder. Total RNA was extracted from the mixture of leaf stem and root tissues using the plant RNA plus Reagent (Takara, Dalian, China) and treated with a PrimeScript RT reagent Kit with gDNA Eraser (Takara, Dalian, China) to eliminate DNA according to the manufacturer's protocols (https://www.takarabiomed.com.cn/, accessed on 17 August 2022). The total RNA samples were stored at −80 °C.

### 2.3. Cloning of StoWRKY6 and Bioinformatics Analysis

Analyses of the transcriptome database of *Solanum torvum* infected by *Verticillium dahlia* candidate WRKY sequences (Stor_Unigene_30993) were obtained (https://www.ncbi.nlm.nih.gov/sra/SRX9839039, accessed on 13 June 2022). The full-length cDNA of *StoWRKY6* was amplified using a pair of *StoWRKY6*-specific primers P1 (gggATGGACAAAGGATGGGGTC) and P2 (cggatccgATTATTATTGCTGGAGGGAC). Similarity analyses of the amino acid and nucleotide sequences were compared using the BLAST program at the NCBI GenBank database (http://www.ncbi.nlm.nih.gov/BLAST/, accessed on 15 June 2022). Plant WRKY protein sequences were also obtained from NCBI GenBank. Sequence alignment was performed using ClustalX (version 2.0.8) (Dublin, Ireland) and a phylogenetic tree was generated by a neighbor-joining algorithm with a p-distance method using MEGA version 6.0(Philadelphia, Pennsylvania) [20]. A bootstrap statistical analysis was performed with 1000 replicates to test the phylogeny. Primers were designed according to the eggplant genomic data, and the 2000 bp promoter sequence ofStoWRKY6 was cloned using eggplant genomic DNA as a template. The sequence was analyzed by plantpan 3.0 software (http://plantpan.itps.ncku.edu.tw/index.html, accessed on 20 June 2022).

### 2.4. Construction of VIGS Vector and Fungi Inoculation

Tobacco rattle virus (TRV)-based vectors (pTRV1 and pTRV2) and *Agrobacterium tumefaciens* strain GV3101 were used for the VIGS assay. *StoWRKY6* was amplified by PCR using P3 (ctctagagATGGACAAAGGATGGGGTC) and P4 (cggatccgGGCACACTTGCAAGTTGAGC). BamHI/XbaI was used to digest the PCR fragments of *StoWRKY6* and the end products were inserted into the pTRV2 plasmid. The construct was then introduced into *A. tumefaciens* by electroporation.

For Agro-infiltration, *Agrobacterium* cultures containing pTRV2: StoWRKY6 and pTRV1 were mixed 1:1 and then infiltrated into two fully expanded cotyledons of 10-day-old *Solanum torvum* seedlings by a needle-less syringe. The bacterial solution was injected into the back of two cotyledons to fill the whole cotyledon. The plants were watered before and after infection to prevent plants from wilting due to lack of water. The expression level of *StoWRKY6* was checked by qRT-PCR in pTRV2: StoWRKY6 infiltrated plants when the albino phenotype appeared on the true leaves of PDS-silencing plants. *StoWRKY6* silencing plants were subjected to *Verticillium dahliae* inoculation by the root-dipping method described above.

### 2.5. Subcellular Localization Analysis

The full-length cDNA of *StoWRKY6* was amplified and cloned into the pBinGFP4 vector to yield the GFP-StoWRKY6 construct. The pBinGFP4- GFP- StoWRKY6 construct and pBinGFP4 empty vector (as a control) were transformed into *A. tumefaciens* strain GV3101 by electroporation, separately. Agrobacteria carrying pBinGFP4- GFP-StoWRKY6 or pBinGFP4 were grown at 28 °C in LB media (50 μg/mL kanamycin and 50 μg/mL rifampicin), pelleted by centrifugation, and then resuspended in infiltration buffer (10 mM $MgCl_2$, 10 mM MES and 150 μM acetosyringone) to $OD_{600}$ = 1. Agrobacterial suspensions were infiltrated into the leaves of *N. benthamiana* using 1 mL syringes without needles. Finally, 48 h after infiltration, GFP-fluorescence was observed by a confocal laser scanning microscope.

### 2.6. Quantitative Real-Time PCR (qRT-PCR) Analysis of Gene Expression

For tissue-specific expression analysis, the root, stem, and leaf at five-leaf stages were collected separately for RNA isolation. Expressions of *StoWRKY6* in the root, stem, and leaf were analyzed by qRT-PCR. The silencing efficiency of *StoWRKY6* was evaluated by a reduction in the mRNA transcript level in TRV-*StoWRKY6* infiltrated plants, and the transcript level of *StoWRKY6* was analyzed by qRT-PCR. The root tissue of four true leaf seedlings was collected at 0, 12, 24, 36, 48, and 72 h post-inoculation (hpi). The total RNA was prepared as described above. *β-Actin* was used as a reference gene with primers of β-Actin-F (ACTGAGGCACCCCTTAATCCC) and β-Actin-R (ACACCATCACCAGAGTCCAACAC). P5 (TCAAGGGATAAGACAACTGGC) and P6 (GGTTGCTTCTGTTGATTGCTC) primers were used to detect *StoWRKY6*. *StoWRKY6* transient expressed leaves and the control group of *N. benthamiana* were collected at 24 h and 48 h after inoculation. The total RNA was prepared as described above. The relative expression was calculated using the $2^{-\Delta\Delta CT}$ method as described previously.

### 2.7. Statistical Analysis

Each determination was repeated at least three times in all of the experiments. Values in the columns are the means of the experiments conducted in triplicate and standard errors of the means are indicated in the bars. Duncan's multiple range tests were performed using one-way analysis of variance (ANOVA) in SPSS version 17.0 (Chicago, Illinois) for Windows software and a $p$-value < 0.05 was considered statistically significant.

## 3. Results

### 3.1. Cloning and Characterization of StoWRKY6

The up-regulated expression of the *WRKY* transcription factor was selected from transcriptome sequencing data of *Solanum torvum* infected by Verticillium wilt. The full-length gene was polymerized and entitled *StoWRKY6*. The full-length cDNA of *StoWRKY6* is 1653 bp with an open reading frame of 1650 bp, which encodes a protein of 535 residues. The StoWRKY6 protein contains one WRKY domain (Figure 1a). The core conservative domain is located at 304–363 (Figure 1b). Phylogenetic tree analysis revealed that StoWRKY6 shows the highest identity to *Solanum melongena* SmWRKY6 (Figure 1c), which is a positive regulator of *R. solanacearum* resistance and heat-stress tolerance. This outcome implies

a role of *StoWRKY6* in regulating disease resistance responses in eggplant. To further understand the potential function of the gene, we cloned the first 2000 bp UTR fragment and analyzed the conserved motifs. Some valuable domains were found in this region, such as the CGTCA-motif related to jasmonic acid response, W-box domain related to WRKY transcription factor interaction, ABRE-motif related to abscisic acid response, and ERE-motif related to ethylene response.

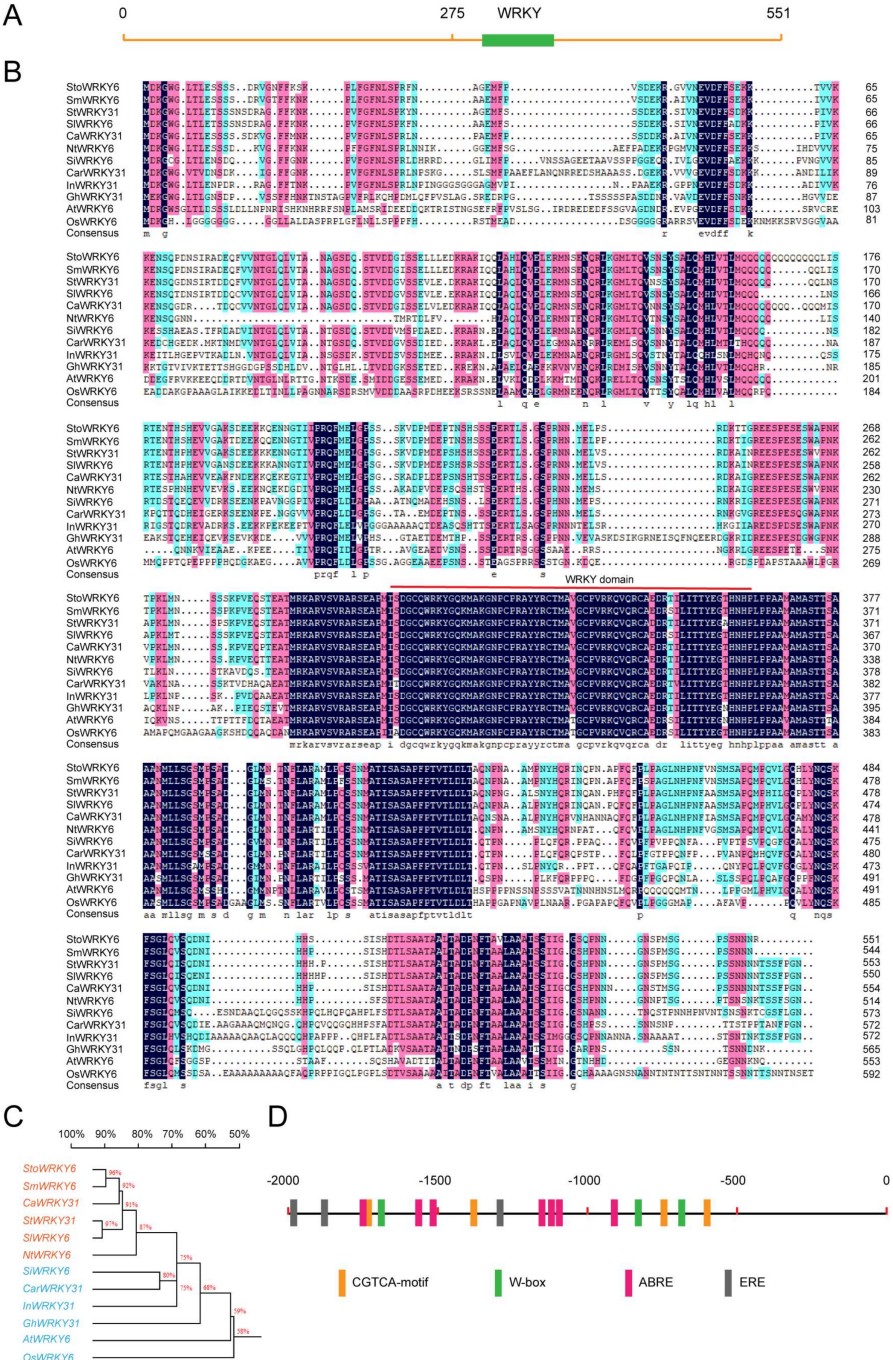

**Figure 1.** Sequence alignment and phylogenetic tree analysis of StoWRKY6 with other plant WRKY proteins. (**A**) Analysis of the core conservative domain of StoWRKY6. (**B**) Alignment of StoWRKY6 with SmWRKY6, CaWRKY31, SlWRKW31, SlWRKY6, NtWRKY6, SlWRKY6, CarWRKY31, In-WRKY31, GhWRKY31, AtWRKY6, and OsWRKY6. (**C**) Phylogenetic tree analysis of StoWRKY6 with other plant WRKY proteins. The phylogenetic tree was constructed by the neighbor-joining method using the MEGA program (version 6.05). (**D**) Analysis of the *StoWRKY6* promoter region.

### 3.2. StoWRKY6 Is a Nuclear Protein

To determine the subcellular localization of StoWRKY6, *N. benthamiana* plants were selected as a carrier, and a GFP- StoWRKY6 fusion construct was generated and transferred into the leaves of *N. benthamiana*. When transiently expressed in tobacco, the GFP-StoWRKY6 fusion protein was localized exclusively in the nucleus, whereas the GFP4 fluorescence was observed throughout the cytoplasm and nucleus without specific compartment localization (Figure 2). This result indicates that the StoWRKY6 is localized in the nucleus of cells, as is expected for a TF.

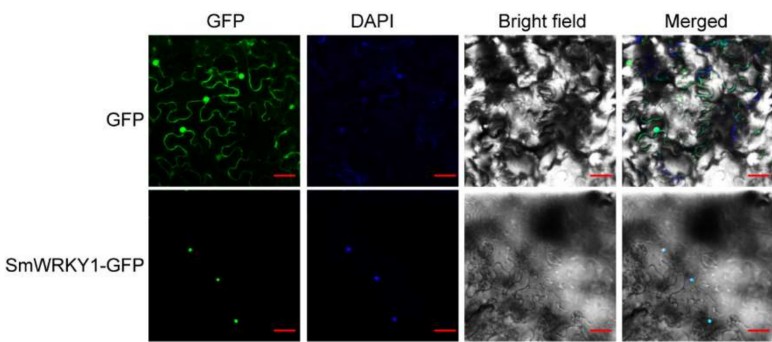

**Figure 2.** Subcellular localization of StoWRKY6 when transiently expressed in *N. benthamiana* leaves. The GFP-StoWRKY6 and GFP4 constructs were transferred into *N. benthamiana* leaves separately through agro-infiltration and the green fluorescence of the GFP4 was observed under confocal laser microscopy. Bar = 40 μM.

### 3.3. Tissue-Specific Expression Analysis and Expression of StoWRKY6 in Response to Verticillium dahliae

To further clarify the potential functions of the *StoWRKY6* gene, the expression patterns in different organs of *Solanum torvum* were first examined by qRT-PCR. As shown in Figure 3a, the highest expression level of *StoWRKY6* was detected in the root, whereas it was low in the leaf and stem.

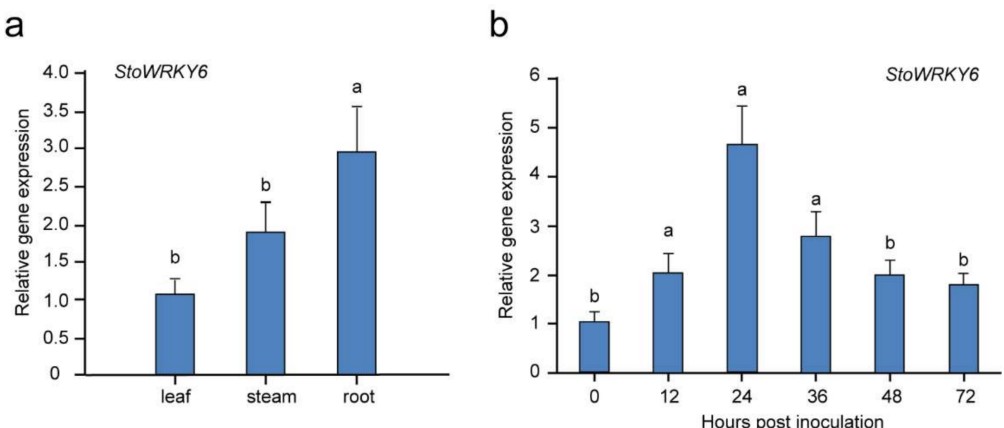

**Figure 3.** Tissue-specific expression and expression of *StoWRKY6* in response to *Verticillium dahliae* infection. (**a**) Tissue-specific expression of *StoWRKY6* in root, stem, and leaf. (**b**) Expression patterns of *StoWRKY6* in response to *Verticillium dahliae* inoculation. The leaves were collected at 0, 12, 24, 36, 48, and 72 hpi, respectively. Different lowercase letters indicate significant differences based on the Fisher's protected LSD test ($p < 0.05$).

To explore the possible involvement of *StoWRKY6* in the *Solanum torvum* disease resistance response, we analyzed the expression dynamics of *StoWRKY6* in response to

infections of *Verticillium dahliae*. As shown in Figure 3b, the expression level of *StoWRKY6* was significantly up-regulated 12 h after inoculation and reached the peak at 24 h, which showed an increase of 4.65-fold compared to 0 h hpi. Subsequently, the expression remained significantly high at 36 h, but this significant difference disappeared after 48 h of inoculation.

### 3.4. Silencing of StoWRKY6 Affects the Defense Response against Verticillium dahliae

We employed the TRV-vector-based VIGS system to test the role of *StoWRKY6* in *Solanum torvum* resistance to Verticillium wilt. The silencing effect was monitored with PDS, a gene involved in chloroplast development. At approximately two weeks after manual infiltration, silencing of PDS led to an albino phenotype for newly developed true leaves (Figure 4a). This demonstrated that the VIGS approach was successful. Further examination of *StoWRKY6* showed that its expression was effectively reduced in *StoWRKY6* silenced plants, suggesting that the gene had been effectively silenced (Figure 4b). At 25 dpi, no apparent disease symptoms were observed in the control tissues, whereas the leaves from *StoWRKY6* silenced plants were yellowish, distorted, and shrunken (Figure 4c). Compared with the control, the disease index of *StoWRKY6* silenced plants was higher (Figure 4d). To further verify the pathogen-infected plants, we cultured the stems of the treated *StoWRKY6* silenced plants on a PDA medium. After 5 days, hyphae appeared on both ends of stem segments of WRKY silenced plants, but not in the control treatment (Figure 4e). The existence of *Verticillium dahlia* was also verified by qRT-PCR (Figure 4f).

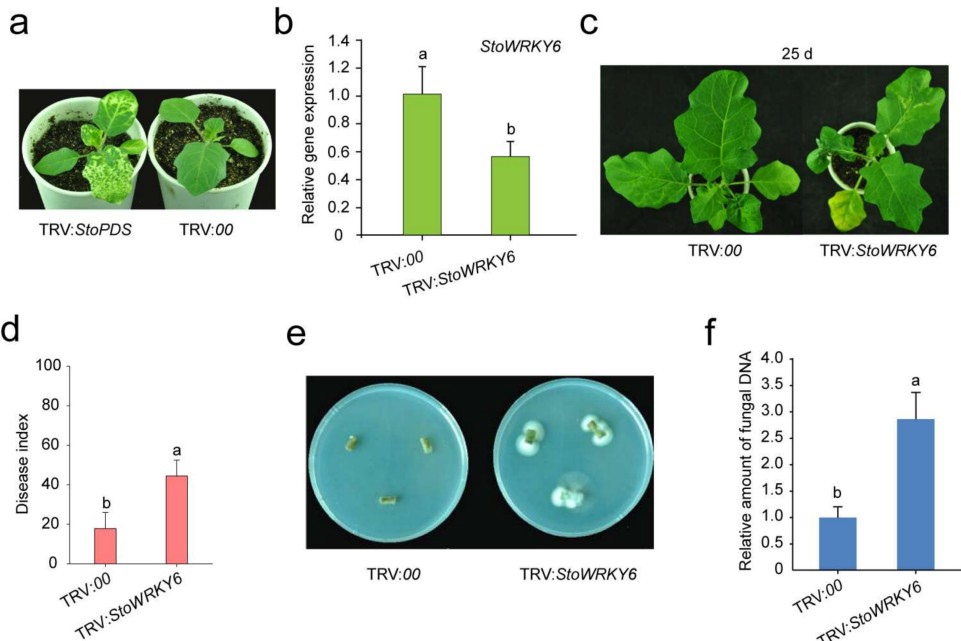

**Figure 4.** Silencing of *StoWRKY6* seriously compromises Verticillium wilt resistance in *Solanum torvum*. (**a**) PDS as a visual marker for silencing efficiency. (**b**) Relative expression level in the control and *StoWRKY6* silenced plants 15 d after manual infiltration. (**c**) Disease symptoms induced by *V. dahlia* on the control and *StoWRKY6* silenced plants at 25 dpi. (**d**) The relative expression level of *V. dahlia* in the control and *StoWRKY6* silenced plants at 25 dpi. (**e**) Recovery test of the pathogen from stem segments of *StoWRKY6* silenced plants. (**f**) Detection of pathogenic bacteria in stems by qRT-PCR. Different lowercase letters indicate significant differences based on the Fisher's protected LSD test ($p < 0.05$).

### 3.5. Identification of Disease Resistance after Transient Expression of StoWRKY6 in Tobacco

The *Agrobacterium tumefaciens* mediated transfection was used to express GFP-StoWRKY6 transiently in *N. benthamiana*, whereas pBinGFP4 was used as a control to observe the lesion size under UV light after 36 h of inoculation. Obvious necrotic plaques appeared on

the tobacco leaves with transient expression of *StoWRKY6*, whereas the control group was normal (Figure 5a). After 24 and 48 h of treatment, the electrical conductivity of *StoWRKY6* transient expressed leaves was 50.25 μS/cm and 78.86 μS/cm, and that of the control was 23.68 μS/cm and 32.58 μS, respectively (Figure 5b). The relative expression level of *StoWRKY6* in agro-transfected tissues proved that the *StoWRKY6* was successfully expressed (Figure 5c). In addition, we also checked the expression of disease-related genes to determine whether they are affected by the expression of target genes. *NbPR2a* and *NbPR2b*, two genes related to disease resistance, were up-regulated significantly in tobacco leaves with transient expression of *StoWRKY6* (Figure 5d). Taken together, *StoWRKY6* plays a significant role in resistance to *V. dahlia* infection.

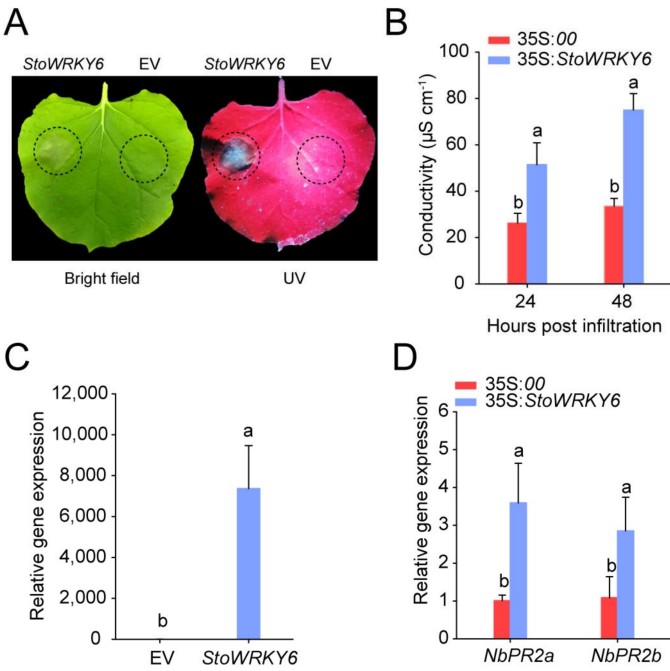

**Figure 5.** *StoWRKY6* stimulated tobacco anaphylactic reaction. (**A**) Lesion phenotype of *N.benthamiana* leaves infection with *Verticillium dahliae.* (**B**) Electrical conductivity of tobacco leaves at 24 h and 48 h after transient expression of *StoWRKY6*. (**C**) Relative expression of *StoWRKY6.* (**D**) Relative expression of disease resistance-related genes. Different lowercase letters indicate significant differences based on the Fisher's protected LSD test ($p < 0.05$).

## 4. Discussion

Many members of the WRKY family are involved in the regulation of plant disease resistance and may be related to each other at the transcriptional and post-transcriptional levels, forming a complex WRKY network that plays an important role in the rapid and successful regulation of plant response to pathogens. As an important solanaceous vegetable, eggplant has a different evolutionary environment from rice, Arabidopsis, and other plants, and the disease problem is relatively prominent. However, the research on the regulatory role of the eggplant WRKY family is limited compared with rice, Arabidopsis, and other plants. It has been reported that several plant WRKY transcription factors were tissue-specific [21]. In our experiment, the *StoWRKY6* was predominantly expressed in the roots, but at relatively low levels in leaves and stems. qRT-PCR analysis was used to investigate the expression patterns of *StoWRKY6*. The expression levels were up-regulated after Verticillium wilt infection. In addition to the role in the biotic stress response, the WRKYs have also been implicated in abiotic stress responses [22]. In our study, we found that the silencing of *StoWRKY6* could affect the phenotype of disease caused by Verticillium wilt and the biomass of *Verticillium dahlia* was higher. Moreover, transient expression of *StoWRKY6* in *N. benthamiana* enhanced diseases caused by *Phytophthora nicotianae*, which

suggested that *StoWRKY6* is involved in the biotic stress response. Together, these data suggest that *StoWRKY6* is required for disease resistance against Verticillium wilt. The plant response to biotic and abiotic stresses is a very complex trait involving different signaling pathways and gene regulation. We demonstrated that *StoWRKY6* is a positive regulatory component of eggplant against Verticillium wilt. The underlying mechanism of the complicated regulation of the biotic stress response by Verticillium wilt requires further study.

VIGS technology is convenient in performing gene function verification for plants not needing to undergo a stable process of genetic transformation, especially for crops such as eggplant that need to undergo explant dedifferentiation and regeneration. Currently, tobacco embrittlement virus (TRV)-induced gene silencing (TRV-VIGS) is a widely used silencing system [23]. The cotyledons of young eggplant seedlings were infected by the injection compression method at 20–25 °C which effectively reduced the expression of target gene transcription level in the leaves and provided a technical platform for functional genomics in eggplant. Factors affecting the efficiency of VIGS include environmental conditions, the length of the target gene, the mode of infection, and the age of inoculated seedlings. In this study, when the diurnal temperature was controlled at 20–25 °C and the relative humidity was 60%–75%, the inserted fragment infected the cotyledons of eggplant seedlings at about 200 bp, and the silencing effect was significant. This is consistent with previous research results in *Arabidopsis thaliana* and *Nicotiana tobacco* [24]. Liu et al. (2012) infected 4-week-old eggplant seedlings with a high-pressure spray gun method and leaf injection method, and results showed that the high-pressure spray gun was more effective in silencing [25]. This may be due to the obvious fibrosis of true leaves, the low pressure of the injection compression method, and that the bacteria liquid could not easily penetrate, resulting in a small amount of virus in the leaves and an unobvious silencing effect. However, the high-pressure spray gun method can ensure the amount of virus infected in the plant and can produce an obvious phenotype. The cotyledon of young seedlings was used as the infected object in this study, and the cotyledon injection method achieved an ideal silencing effect. The results showed that the VIGS infection silencing effect of eggplant was related to infection method and eggplant plant size. The high-pressure spray gun method was suitable for infecting older plants, and the leaf injection method was suitable for infecting eggplant cotyledon seedlings. In potato, poplar, cotton, and other crops, genotypes have an impact on the silencing efficiency of VIGS [26]. At present, VIGS has been successfully applied in many plants, such as tobacco, Arabidopsis, rice, and others, but there are relatively few reports in the study of eggplant functional genes, therefore, this study shows that using VIGS to verify gene function in eggplant has high feasibility.

The function of WRKY6 has been reported in other plant disease resistance studies. For example, based on transient expression analysis, *OsWRKY6* positively regulates *OsPR10a* expression through physical interaction with the w-box element. Therefore, salicylic acid-induced *OsWRKY6* was identified as a positive regulator of the rice disease-related defense gene *OsPR10a*. Rice lines overexpressing *WRKY6* showed stronger resistance to pathogens, while RNAi interfering lines showed weaker resistance. This indicates that *OsWRKY6* is a positive regulator of the plant defense response [27]. WRKY6 can respond to *Botrytis cinerea*, *Pseudomonas syringae*, *Phytophthora infestans,* and *Erysiphe orontii*, indicating the diversity of its regulatory functions in plant stress response. In addition, the WRKY transcription factor can play a role in the regulation of defense response in the early stage of disease resistance, so WRKY6 will respond quickly, and the expression level will increase when the pathogen is treated [28]. In conclusion, WRKY6 can respond to biological stress in different species. In this study, the result shows that StoWRKY6 can respond to the infection of *Verticillium dahliae* in eggplant and tobacco, indicating that the resistance of WRKY6 among different species is conservative.

The transcriptional regulation of genes in host plants is a critical step that activates inducible defense responses upon pathogen infection. Even though extensive studies have

been performed and many WRKYs have been implicated in the biotic and abiotic stress response in model plant species, such as Arabidopsis and rice [29–32], little is known about the biological function of eggplant WRKYs. In this study, we demonstrated the StoWRKY6 function as a positive regulator in defense response against Verticillium wilt. As a large TF family, members of the WRKY family probably participate in diverse biotic and abiotic stress responses. The findings presented in this study provide a previously uncharacterized member of the eggplant WRKY family with well-defined functions in biotic stress responses. These results also provide a basis for further study on the signal transduction of stress response genes, elucidation of the relationship between the regulation of stress response genes, improvement of plant disease resistance, and molecular mechanism of eggplant resistance to Verticillium wilt.

**Author Contributions:** All authors contributed to the experimental design. X.Y. designed the experiments. Y.Z. performed the experiments. L.S. and L.L. analyzed the data. Y.Z. and X.Y. wrote the paper. All authors have read and agreed to the published version of the manuscript.

**Funding:** The work presented here was supported by the National Natural Science Foundation of China (Grant No. NSFC31972395). The funder had no role in the study design, data analysis, or preparation of the manuscript.

**Data Availability Statement:** All data included in this study are available upon request by contact with the corresponding author.

**Conflicts of Interest:** The authors declare no conflict of interest.

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
