# Peer review of "The Solanum torvum Transcription Factor StoWRKY6 Mediates Resistance against Verticillium Wilt"

_agronomy, doi:10.3390/agronomy12081977_

Round 1

Reviewer 1 Report

Carefully revise the manuscript.

- line 80: ("Robatzek ... ") reference number missing. 

- line 88: "Yang et al. (2014)" - Follow guide for the MDPI in-text referencing; Yang et al. [ref. number]

- I noticed some typos (e.g. lines 83-84) "... in continuous β After aminobutyric..."

- line 91 "In addition, Shao et al. (years)..." 

- in general: Latin names should be italic.

2.2. Extraction of total RNA

- Im missing more exact procedure descriptions or at least references by which authors followed; e.g. did you ground the tissue, and if so, how?

-line 125 : "Plant WRKY protein sequences were also extracted  obtained from NCBI GenBank."

- line 127 - MEGA version 6.05. (reference needed).

-line 131 in-text reference "[20]" probably incorrect as the study describes over-expression procedure not VIGS.

- line 136 "Agrobacterium" italic

- lines 183 - 187 - you mention cloning and analysis of promoter region. Very interesting findings. However, please briefly describe or explain this procedure in the section "2.3. Cloning of StoWRKY6 and bioinformatics analysis" as this is missing.

Figure 2. captions: "Scale bars = ..."

In section "3.3 Tissue-specific expression analysis and expression of StoWRKY6 in response to Verticillium dahliae" explain figure 3b in text in more detail. You observed first significant increase between 0 and 12 hpi, however you mention only highest relative gene expression at 24 hpi and its comparison to 0 hpi. I would expect a more detailed description, e.g., "The first significant increase was observed at 12 hpi, and the highest relative expression was observed at 24 hpi," etc...

Figure 3: In the caption, explain what the letters mean - e.g. "different letters denote statistically significant differences...". Also indicate what statistical test you used and the level of significance.

At line 232 you say "At 25 dpi..." and also in caption of figure 4c ("25 dpi"), however in figure 4c you have "10 d" - what this means? 10 days or something else?

Explain letters at graphs of figure 4c, 4d, 4f. Same for figure 5b, 5c, 5d

In section "3.5. Identification of disease resistance after transient expression of StoWRKY6 in tobacco" you present the results of expression analysis of NbPR2a and NbPR2b genes. Please describe the procedures of RNA isolation from N. benthamiana in section "2.2. Extraction of total RNA", its qPCR analysis in section "2.6. Quantitative real-time PCR (qRT-PCR) analysis of gene expression" and statistical analysis for those two genes in section "2.7"

Discussion can be improved. You are stating that : "In this study, when the diurnal temperature is controlled at 20–25℃ and the relative humidity was 60%–75%, the inserted fragment infects the cotyledons of eggplant 300 seedlings at about 200 bp, and the silencing effect is significant.". However, you have not mentiond observing this parameters in the Material and methods section. More detail discussion on your results is recommended. 

Reviewer 2 Report

Dear authors, the manuscript reports the transcription factor StoWRKY6 that mediates resistance against verticillium wilt. The manuscript is well written, and the study demonstrates the StoWRKY6 function as a positive regulator in defense response against Verticillium wilt. However, some minor changes are suggested and reported in the comments.

Regards

Introduction

Comment 1: Line 26

Substitute with: Eggplant (Solanum melongena) is one of the most important vegetable crops in the world.

Comment 2: line 31-32

Please substitute with at present, many studies have been conducted on the resistance mechanism of eggplant to verticillium wilt.

Comment 3: line 33 please provide references on the studies already conducted. 

Comment 4: Line 49:  please provide a reference.

Comment 5:  Line 57: same, please provide a reference.

Comment 6: line 80 please correct the citation and add the year, Robatzek et al. 2001

Line 91: shao et al. same the year is missing in the text (2014)

 Material and methods

Comment 7: Line 104: how many plants were used of eggplant and Nicotiana benthamiana?

Comment 8: Line 108: the fungus was cultured for how many days, please add in the text

Comment 9: I suggest adding more details about the dipping method

Comment 10: Line 114: stem is not spelled correctly

Discussion:

Line 276 : qRT-PCR 

Author Response

This manuscript is a resubmission of an earlier submission. The following is a list of the peer review reports and author responses from that submission.